# Benefits of Physical Activity and Its Associations with Resilience, Emotional Intelligence, and Psychological Distress in University Students from Southern Spain

**DOI:** 10.3390/ijerph17124474

**Published:** 2020-06-22

**Authors:** Silvia San Román-Mata, Pilar Puertas-Molero, José Luis Ubago-Jiménez, Gabriel González-Valero

**Affiliations:** 1Department of Nursing, University of Granada, 18071 Granada, Spain; silviasanroman@ugr.es; 2Department of Didactics of Musical, Artistic and Corporal Expression, University of Granada, 18071 Granada, Spain; pilarpuertasmolero@gmail.com (P.P.-M.); ggvalero@ugr.es (G.G.-V.)

**Keywords:** resilience, emotional intelligence, psychological distress, students

## Abstract

This is a descriptive and cross-sectional study in a sample of 1095 university students from southern Spain. The aim was to identify the frequency of health-fulfilling physical activity engagement reported by participants. Sufficient physical activity was categorized according to whether participants ‘achieved minimum recommendations’ (≥150 min of moderate physical activity) or ‘did not achieve minimum recommendations’ (≤150 min of moderate physical activity). Participants were further categorized as: inactive (does not engage in physical activity or sport), engaging in physical activity that is not beneficial to health (≤300 min of moderate physical activity per week) and engaging in physical activity that is beneficial to health (≥300 min of moderate physical activity per week). Possible relationships with psychosocial factors and perceived psychological distress were explored. An ad hoc questionnaire was used to record the time in minutes of physical activity engagement per week. The Connor–Davidson Resilience Scale, the Trait Meta-mood Scale, and Kessler Psychological Distress Scale were also administered. Statically significant differences are shown between the three examined groups: physical inactivity and non-beneficial physical activity; physical inactivity and beneficial physical activity, and; non-beneficial physical activity and beneficial physical activity. Positive and direct correlations were seen with respect to resilience and understanding, and emotional regulation, in addition to negative associations with respect to psychological distress. In conclusion, the more individuals engage in beneficial physical activity, the greater their resilience and emotional management, and the lower their rates of psychological distress.

## 1. Introduction

Regularly engaging in physical activity is considered a healthy habit and a protective factor against diseases and harmful risk behaviors. It provides multiple benefits such as improving self-esteem and body image, decreasing stress and nervous tension, improving motor balance, and favoring social relatedness [1,2,3]. In this way, individuals can relate and socialize with their peers simply by going to the gym, or belonging to a team or sports club.

Authors such as Polo-Gallardo, Cobos, Mendinueta-Martínez, and Acosta [4] explain that physical activity can be considered as a type of non-pharmacological therapy. There is no doubt that physical activity engagement promotes psychological well-being in various types of individuals [5,6].

World Health Organization (WHO) physical activity recommendations [7] for adults aged from 18 to 64 years of age, suggest a minimum of 150 min per week of moderate activity [8,9,10]. There is a direct relationship between physical activity and cardiorespiratory health, but meaningful risk reductions are achieved from 150 min of moderate or intense exercise a week. In cases where physical activity engagement increases to 300 min a week or above, additional health benefits are reported.

Despite knowledge of the many benefits associated with physical activity engagement, it seems that engagement in physical exercise decreases during adolescence and youth [11,12].

Likewise, Práxedes, Sevil, Moreno, del Villar, and García-González [13] state that the university stage is a critical period. Nonetheless, they found that more than half of the university students included in their study failed to reach the minimum recommendations for physical activity. Our previous research also highlights issues related with sedentary habits in adolescents [14]. Some authors think that one of the causal agents is due to the predisposition of young people towards other more sedentary leisure activities such as the use of new information technologies and the internet [15,16,17]. In order to promote healthy behavior or prevent harmful risk behavior, psychological processes inherent to each stage of change must be attended to.

In this way, resilience and emotional intelligence are the most relevant psychosocial or intrinsic factors involved in the teaching-learning processes, with others being the development of activities and socialization processes. Resilience can be defined as the set of intrinsic factors which characterize individuals involved in the process of overcoming adversity, consequently emerging stronger from this process [18,19]. In addition, resilient individuals tend to be dynamic and capable of learning [20]. In this way, resilience is seen as an associated factor promoting psychological well-being in the population [21,22].

Emotional intelligence is a current subject of great interest [23]. At the start of its theoretical development, Goleman [24] explained that individuals who control their feelings, and recognize and interpret them, show certain advantages over people who do not master their emotional life [25]. For this reason, emotional intelligence may be defined as the ability or competence to solve problems derived from emotions, that is, being able to perceive, understand, and regulate emotions. This occurs both in relation to one’s own emotions and those of other people around them [26,27]. It is dynamic in nature [28], changing over time as a result of personal development and growth.

The university period is considered to be a time of evolution and great change in young people. During this stage, individuals first encounter acute and chronic risky situations. In addition, transition to university entails a large step-up as far as academic level is concerned, whilst the emergence of and need to manage new social networks can cause stress and alter the psychological well-being of university students. This study aimed to uncover whether the frequency of physical activity engagement reported by participants can be considered to be healthy, whilst, at the same time examining its potential relationships with psychosocial factors and perceived psychological distress. The objectives of the present study are as follows: Identify the relationship between the adequacy of physical activity engagement, psychosocial parameters (resilience and emotional intelligence), and psychological discomfort. This will uncover associations according to whether individuals are classified as being inactive, engaging in physical activity but below levels considered to be beneficial, and engaging in beneficial physical activity. Such relations will be examined as a function of psychosocial factors (resilience and emotional intelligence) and psychological distress. Correlational analysis is performed between those who do not meet minimum physical activity recommendations and those who do, according to psychosocial factors (resilience and emotional intelligence) and psychological distress

## 2. Materials and Methods

### 2.1. Participants

1095 university students from Andalucía and Melilla participated in this descriptive and cross-sectional study. A total of 23.5% of participating universities came from eastern Andalusia (Sevilla, Huelva, and Malaga), 24.6% from western Andalusia (Granada and Almeria), and 37% came from the Melilla University Campus. Participants were undertaking the following academic degrees: Health Science (56%), Educational Science (27.2%), Engineering (7.5%), Law (7.4%), and a Master’s degree (1.9%). The sample was represented by 67.9 % females and 32.1% males, will all participants being aged between 17 and 54 years (M = 21.4; SD = 4.6). Inclusion criteria for the present study meant that, in order to be eligible, potential participants must have been studying at an Andalusian University at the time of data collection, be proficient in the management of new technologies, and have an internet connection.

### 2.2. Variables and Instruments

#### 2.2.1. Ad-Hoc Questionnaire

The age and gender of individuals were recorded in this self-registration sheet. In order to avoid further bias, questions were also included about the time in minutes of moderate or intense physical activity engagement each week and relative personal ability. In the case of the latter, participants responded on a scale which ranged from 0 (does not engage in physical activity or sport) to 10 (above possibilities). Scores of 5 or 6 equate to moderate physical activity Once this data was collected, WHO [7] physical activity recommendations provided a framework to calculate whether minimum recommended physical activity levels were being achieved. Categories corresponded to whether individuals “met minimum levels” (≥150 min of moderate physical activity) or “did not meet minimum levels” (≤150 min of moderate physical activity). Further, participants were also categorized as: inactive (does not engage in physical activity or sport), engages in physical activity but at a level that is not beneficial to health (≤300 min of moderate physical activity per week) and engages in physical activity that is beneficial to health (≥300 min of moderate physical activity per week).

#### 2.2.2. Connor–Davidson Resilience Scale

This questionnaire is composed of 25 items which are rated using a four-point Likert scales (1 “almost never” and 4 “almost always”). Items were summed to determine general resilience, with this then being grouped into five dimensions: locus of control and commitment (LCC), challenge of action-oriented behavior (CAOB), self-efficacy and resistance to discomfort (SRD), optimism and adaptation to stressful situations (OASS), and spirituality (ES) [29].The internal reliability coefficient obtained for this tool was α = 0.944, whilst LCC was α = 0.763, CAOB was α = 0.650, SRD was α = 0.905, OASS was α = 0.775, and ES was α = 0.498.

#### 2.2.3. TMMS-24 Questionnaire

The TMMS-24 questionnaire, based on the trait meta-mood scale developed by Mayer and Salovey [30]. This questionnaire is composed of 24 items which are rated on a five-point Likert scale. All items were summed to determine the dimensions of emotional perception, understanding, and regulation. Examination of internal reliability produced coefficients of α = 0.916 for emotional perception (EIP), α = 0.918 for emotional understanding (EIU), and α = 0.891 for emotional regulation (EIR).

#### 2.2.4. Questionnaire of Psychological Distress

The original version of this tool (Kessler Psychological Distress Scale K10 [31]) has been validated into Spanish by Alonso, Herdman, Pinto, and Vilagut [32]. It is composed of 10 items and produced an internal reliability coefficient of α = 0.895.

### 2.3. Procedure

Collaboration was requested from various universities across Andalusia and the Melilla University campus through an information letter drawn up by the University of Granada’s research team. The nature and objectives of the research were specified, and informed consent was requested from participants. Data collection was carried out during university hours, under supervision of the researchers and teachers of the center. This ensured correct completion of the instruments and enabled doubts to be answered. A total of 59 questionnaires were eliminated due to incorrect completion. The planned procedure was approved by the Research Ethics Committee of the University of Granada (Spain) and respected the ethical principles proposed in the Declaration of Helsinki, ensuring the anonymity and confidentiality of the data.

### 2.4. Statistical Analysis

A descriptive analysis was conducted to determine participants’ characteristics. Means (M), standard deviations (DT), and frequencies (%) were used as basic descriptives. The Student’s *t*-test for independent samples was used to establish relationships between variables, alongside Pearson’s bivariate correlations. Significance was established at the level of *p* < 0.05 and *p* < 0.01. Normality and homogeneity of the sample were examined using Kolmogorov–Smirnov’s test. The magnitude of differences, in other words the effect size (ES), was obtained using Cohen’s standardized d [33] with effects being interpreted as null (0–0.2), low (0.20–0.50), moderate (0.50–0.79), or high (≥0.80). The 95% confidence interval (CI) was calculated for each effect size. Data were analyzed using SPSS statistical software version 25.0 (IBM Corp, Armonk, NY, USA)

## 3. Results

Statistically significant differences were found in regards to relationships between physical activity engagement, as defined according to meeting WHO recommendations [7], and the psychosocial parameters of the present study: resilience, emotional intelligence, and psychological distress. In this sense, it can be seen that those who meet minimum requirements for physical activity each week have better averages values of general resilience (M = 3.83 ± 0.706; ES = 0.224) when compared to those engage in insufficient physical activity levels (M = 3.67 ± 0.720). This same pattern is seen to occur in relation to the dimensions of LCC (M = 3.82 ± 0.843; ES = 0.254), CAOB (M = 3.82 ± 1.021; ES = 0.029), SRD (M = 4.00 ± 0.772; ES = 0.246), and OASS (M = 3.85 ± 0.775; ES = 0.283), with better average values being reported by those who meet minimum weekly physical activity targets.

In the same way, the same outcome is seen in relation to emotional intelligence. Again, those who meet minimum physical activity recommendations also obtain better scores for EIU (M = 3.04 ± 0.846; ES = 0.119) and EIR (M = 3.28 ± 0.851; ES = 0.235). This same relation is seen with psychological distress, with those not meeting minimum physical activity recommendations reporting higher levels of PD (M = 2.46 ± 0.725; ES = 0.369) (Table 1).

Likewise, significant differences are found in many of the associations found with the remaining psychological variables examined in the present study when analyzed as a function of physical inactivity, and non-beneficial physical activity and beneficial physical activity engagement.

Firstly, in relation to the total resilience score, statistically significant differences are found when comparing the three groups: physical inactivity and non-beneficial physical activity engagement; physical inactivity and beneficial physical activity engagement, and non-beneficial physical activity and beneficial physical activity engagement (*p* ≤ 0.05). Thus, those who reported being physically inactive also reported lower mean resilience scores (M = 3.58 ± 0.752; ES = 0.183) relative to those who do not engage in beneficial physical activity (M = 3.71 ± 0.693; ES = 0.468) and those who do engage in beneficial physical activity (M = 3.92 ± 0.706; ES = 0.301).

With regards to the dimensions of resilience, namely LCC and OASS, significant differences are found in the three associative models proposed in relation to physical activity engagement (*p* ≤ 0.05). Thus, inactive participants presented the lowest means for both LCC (M = 3.50 ± 0.925; ES = 0.173) and OASS (M = 3.51 ± 0.804; ES = 0.221). Furthermore, those who engaged in non-beneficial physical activity presented higher values for LCC (M = 3.65 ± 0.841; ES = 0.476) and OASS (M = 3.68 ± 0.757; ES = 0.573) when compared to inactive participants, but presented lower values when compared with those who engaged in beneficial physical activity (LCC: M = 3.92 ± 0.848; ES = 0.320 and OASS: M = 3.96 ± 0.770; ES = 0.368).

The same findings are also observed in relation to the SRD dimension, with two of the three proposed associations showing significant differences with respect to physical activity engagement (*p* = 0.00). In this case, the best averages were obtained by those who engaged in beneficial physical activity (M = 4.10 ± 0.759; ES = 0.331).

On the other hand, results of the present study fail to show an association between EIP and beneficial physical activity, although statistically significant differences are found in relation to EIU. This association emerges, specifically, when comparing those who engaged in non-beneficial physical activity and those who engaged in beneficial physical activity (*p* ≤ 0.05). The data shows that those with the best average EIU values are those who perform beneficial physical activity (M = 3.10 ± 0.815; ES = 0.194).

Comparisons involving all possible physical activity engagement groups were statistically significant when examining the EIR dimension (*p* ≤ 0,05). Here, associations were revealed which confirmed that highest values for this variable are held by those who engage in beneficial physical activity (M = 3.34 ± 0.848; ES = 0.210), with lower values pertaining to inactive participants (M = 2.94 ± 0.808; ES = 0.260).

Finally, when considering psychological distress and its possible associations with beneficial physical activity engagement, statistically significant differences are observed in the three established assumptions (‘*p* ≤ 0.05’). In the same way, the lowest psychological distress values are presented by those who engage in beneficial physical activity (M = 2.14 ± 0.672; ES = 0.340), followed by those who engage in physical activity levels that are not considered beneficial (M = 2.38 ± 0.721; ES = 0.569) and, finally, those who do not engage in any activity (M = 2.53 ± 0.702; ES = 0.210) (Table 2)

The correlation between individuals who meet minimum recommended levels of physical activity and specified psychosocial parameters is shown. Firstly, a positive and direct correlation is observed between resilience and emotional understanding (*r* = 0.405), and resilience and emotional regulation. (*r* = 0.334). Furthermore, there is a negative and indirect relationship between resilience and psychological distress (*r* = −0.198).

These correlations are largely repeated across all dimensions of resilience in those who meet minimal physical activity guidelines. In this way, positive and direct correlations between the dimensions are obtained: LCC (*r* = 0.326), CAOB (*r* = 0.255), SRD (*r* = 0.277), OASS (*r* = 0.287), and ES (*r* = 0.115). Similarly, EIR is related with the dimensions of LCC (*r* = 0.326; *r* = 0.326), CAOB (*r* = 0.255; *r* = 157), SRD (*r* = 0.277; *r* = 0.313), OASS (*r* = 0.287; *r* = 0.350), and ES (*r* = 0.115; *r* = 0.170). EIP is also positively and directly correlated with the ES dimension (*r* = 0.112), and shows an indirect and negative correlation with psychological distress in the following dimensions: LCC (*r* = −0.233), CAOB (*r* = −0.187), SRD (*r* = −0.178), OASS (*r* = −178), EIU (*r* = −287), and EIR (*r* = −0.245). Psychological distress is directly and positively correlated only with IEP (*r* = 344).

Regarding those who do not engage in minimum recommended levels of physical activity, positive and direct correlations are found with resilience (*r* = 0.205) and the EIP dimensions: CAOB (*r* = 0.090), SRD (*r* = 0.093), and ES (*r* = 0.162). With regards to EIU and EIR, positive and direct correlations are observed with resilience (*r* = 0.364; *r* = 446) and the following dimensions: LCC (*r* = 0.338; *r* = 0.422), CAOB (*r* = 0.270; *r* = 0.285), SRD (*r* = 0.319; *r* = 0.404), OASS (*r* = 0.348; *r* = 0.442), and ES (*r* = 0.177; *r* = 0.288).

Finally, in the group that fails to achieve beneficial physical activity, negative and indirect correlations are uncovered between psychological distress and resilience (*r* = −0.352), LCC (*r =* −0.290), CAOB(*r* = −0.234), SRD (*r* = −0.227), OASS (*r* = −0.227), EIU (*r* = −0.122), and EIR (*r* = −0.150). In turn, there is a positive and direct correlation between EIP (*r* = 0.436) and psychological distress (Table 3).

## 4. Discussion

The present study aims to establish a current perspective in relation to the frequency of physical activity engagement as a beneficial habit for health in university students from southern Spain, whilst also establishing relationships with psychosocial factors and psychological distress. Results demonstrate the existence of statistically significant differences. In the same way, Chow and Choi [34] obtained a positive correlation between resilience, physical activity, and mental health, with higher resilience and greater physical activity engagement predicting greater psychological wellbeing.

In this sense, it can be seen that university students who engage in minimum recommended amounts of physical activity (150 minutes a week) have better average values for resilience and all its dimensions with respect to those who fail to meet guidelines, with the exception being spirituality. Similar outcomes were reached by Szu-Ying, Heng-Hsin, Li-Ning, Liang-Kung, Ching-Iy, and Yen-Ling, [35] in their study, concluding that greater resilience was associated with higher amounts of physical exercise. These findings are also corroborated by Chacón-Cuberos, Castro-Sánchez, Pérez-Turpin, Olmedo-Moreno, and Zurita-Ortega [36] who, when addressing the second hypothesis proposed by their study, showed greater resilience in university students engaging in more than 180 min of weekly physical activity.

The same pattern occurs with emotional intelligence, with participants who meeting minimum physical activity recommendations obtaining better scores in understanding and emotional regulation when compared to those who do not. Similarly, Acebes-Sánchez, Diez-Vega, Esteban-and Gonzalo Rodriguez-Romo [37] found a significant association between physical activity engagement and emotional intelligence. Likewise, Ubago-Jiménez, González-Valero, Puertas-Molero, and García-Martínez [38] carried out a systematic review of the literature and highlighted that a large number of studies identified an association between physical activity engagement and high levels of emotional intelligence. 

In addition, the present work verifies that university students who meet minimum physical activity recommendations also show lower levels of psychological distress relative to other classmates. This is in accordance with ideas presented by Maganto-Mateo, Peris-Hernández, and Sánchez-Cabrero [39]. In this sense, the greater the amount of physical activity performed, the greater the psychological well-being of the individual. Similarly, in their predictive study, González-Hernández and Ato-Gil [40] uncovered a relationship between physical activity engagement, stress responses, and psychological well-being.

Meeting minimum physical activity recommendations is not the same as engaging in healthy physical activity (300 min per week). When considering these differences, our results again reveal significant differences in resilience, emotional intelligence and psychological distress as a function of the three physical activity groupings: physical inactivity, non-beneficial physical activity, and beneficial physical activity.

Likewise, resilience scores and scores related to dimensions of locus of control, optimism, and self-efficacy, were seen to increase as physical activity engagement increased. Concretely, university students engaging in beneficial physical activity levels had higher mean scores compared to the other students, results corroborated by Szu-Ying et al. [35].

On the other hand, results of the present study do not show a significant association between emotional perception and the practice of beneficial physical activity. However, statistically significant differences are found in terms of understanding and emotional regulation, with university students with better mean values being those also engaged in beneficial physical activity. Those engaged in physical activity levels that are below those deemed to be beneficial show better scores than inactive individuals; thus it can be seen that the more physical activity correlates with greater resilience [37].

In reference to the positive and direct correlation found between resilience, and understanding and emotional regulation, this coincides with other studies such as that conducted by Trigeros, Aguilar-Parra, Cangas, Bermejo, Ferrandiz, and López-Liria [41]. That study revealed a positive association between participating adolescents’ emotional intelligence, resilience, and intention to be physically active.

Similarly, it is found in the present study that higher resistance and higher emotional intelligence predict lower psychological distress. This outcome is also supported by Bunce, Lonsdale, Kings, Childs, and Bennie, [42] in their findings. It could be said that our results find a negative and indirect relationship between resilience, emotional intelligence and psychological distress. The present investigation also coincides with the results of What all, Patterson, Siew, Kay-Lambkin, and Hutchesson [43]. Their linear regression model unveiled the existence of a significant association between low scores of psychological distress and high scores of resilience in Australian university students.

The study of Castillo, Fisher, and Dávila [44] which related emotional intelligence with stress and depression factors, similarly to that done in the present study, also showed inverse relationships, with high levels of emotional intelligence predicting low levels of psychological distress. Thus, happiness may increase as a function of greater ability to understand and regulate emotional changes [45].

The limitations of the present study include the fact that university students do not make up the entire youth population which should considered. This is because many young people do not attend university, instead entering the work setting at an early age. The study is, therefore, not entirely generalizable. The geographical area where the sample was selected could also be considered a limitation. As all data were collected in the South of Spain, it is probable that results in relation to psychological well-being and physical activity engagement could be different in other geographical areas, particularly those with a colder and rainier climate. In addition, previous studies have shown that self-reported levels of physical activity are prone to bias and that participants report engaging in higher levels of physical activity when self-reported measures are used, as opposed to objective measures [46,47,48].

## 5. Conclusions

This study of 1095 students from different universities in Southern Spain confirmed the existence of significant relationships between the meeting minimum physical activity recommendations as defined by WHO and the psychosocial parameters of interest to this study: resilience, emotional intelligence, and psychological distress. It is evident that university students who meet the minimum level of recommended physical activity each week, present better average values for general resilience when compared to those who do not meet recommendations.

The same conclusions can be drawn in relation to emotional intelligence, with university students who meet minimum levels of recommended physical activity also obtaining better scores for emotional understanding and regulation. Likewise, existing relationships with psychological distress are revealed. Again, those failing to meet established physical activity guidelines presented higher levels of discomfort. 

In summary, statically significant differences are shown in relation to three different groupings pertaining to physical activity engagement: physical inactivity and non-beneficial physical activity; physical inactivity and beneficial physical activity; non-beneficial physical activity and beneficial physical activity. Positive and direct correlations were reported with respect to resilience, and understanding and regulation of emotions, in addition to negative associations with respect to psychological distress. Greater resilience breeds better emotional management, whilst engaging in beneficial physical activity is linked to lower rates of psychological distress. 

Physical activity programs should be established and implemented from an early age. They should be inclusive and off community sports activities which are easily accessible to the entire population. Concurrent intervention and development programs targeting resilience and emotion management are also proposed as future directions. Likewise, we suggest the expansion and development of research studies into physical activity engagement and psychosocial factors in university students, such research should include a geographically broader sample.

## Figures and Tables

**Table 1 ijerph-17-04474-t001:** Relationships between the dimensions of resilience, emotional intelligence and psychological distress, as a function of physical activity (MPA) adequacy defined according to physical activity guidelines.

Psychosocial Factors	MPA	M	SD	Levene Test		T-Test		ES (d)	95% CI
				F	Sig.	T	Sig.		
SURE	Yes	3.83	0.70	1.26	0.26	3.73	0.00	0.22	[0.10; 0.34]
	No	3.67	0.72						
LCC	Yes	3.82	0.84	1.10	0.29	4.22	0.00	0.25	[0.13; 0.37]
	No	3.60	0.88						
CAOB	Yes	3.82	1.02	1.47	0.22	0.46	0.64	0.02	[−0.09; 0.14]
	No	3.79	1.05						
SRD	Yes	4.00	0.77	0.01	0.90	3.94	0.00	0.24	[0.12; 0.36]
	No	3.81	0.77						
OASS	Yes	3.85	0.77	0.04	0.82	4.65	0.00	0.28	[0.16; 0.40]
	No	3.63	0.78						
ES	Yes	3.30	0.85	1.62	0.20	−0.50	0.61	0.03	[−0.08; 0.15]
	No	3.33	0.81						
EIP	Yes	3.08	0.87	4.47	0.03	−2.32	0.02	0.14	[0.02; 0.26]
	No	3.21	0.91						
EIU	Yes	3.04	0.84	0.17	0.67	1.75	0.07	0.11	[−0.00; 0.23]
	No	2.94	0.84						
EIR	Yes	3.28	0.85	0.01	0.91	3.71	0.00	0.23	[0.11; 0.35]
	No	3.08	0.85						
SUDIS	Yes	2.20	0.67	5.58	0.01	−6.10	0.00	0.36	[0.24; 0.48]
	No	2.46	0.72						

Note 1. Meets minimum physical activity recommendations (MPA). Note 2. Overall resilience (SURE); locus of control and commitment (LCC); challenge of action-oriented behavior (CAOB); self-efficacy and resistance to discomfort (SRD); optimism and adaptation to stressful situations (OASS); spirituality (ES); emotional perception (EIP); emotional understanding (EIU); emotional regulation (EIR); overall psychological distress (SUDIS).

**Table 2 ijerph-17-04474-t002:** Relationships between the dimensions of resilience, emotional intelligence, and psychological distress, as a function of physical activity sufficiency (inactive, below beneficial physical activity, and beneficial physical activity).

Variable	Category	Situation	M	SD	F	Sig	ES (d)	CI 95%
RE	SURE	IN	3.58	0.75	15.94	*p* ≤ 0.05 ^a, b, c^	0.18 ^a^ 0.46 ^b^ 0.30 ^c^	[0.02; 0.33] [0.29; 0.64] [0.16; 0.44]
		NO	3.71	0.69				
		HEALTHY	3.92	0.70				
	LCC	IN	3.50	0.92	18.19	*p* ≤ 0.05 ^a, b, c^	0.17 ^a^ 0.47 ^b^ 0.32 ^c^	[0.01; 0.32] [0.29; 0.65] [0.17; 0.46]
		NO	3.65	0.84				
		HEALTHY	3.92	0.84				
	CAOB	IN	3.73	1.06	0.72	*p* ≥ 0.05	NP	NP
		NO	3.83	1.02				
		HEALTHY	3.82	1.03				
	SRD	IN	3.73	0.80	16.63	*p* = 0.00 ^b, c^	0.47 ^b^ 0.33 ^c^	[0.297; 0.651] [0.189; 0.472]
		NO	3.85	0.75				
		HEALTHY	4.10	0.75				
	OASS	IN	3.51	0.80	22.78	*p* ≤ 0.05 ^a, b, c^	0.22 ^a^ 0.57 ^b^ 0.36 ^c^	[0.06; 0.37] [0.39; 0.75] [0.22; 0.50]
		NO	3.68	0.75				
		HEALTHY	3.96	0.77				
	ES	IN	3.24	0.79	1.16	*p* ≥ 0.05	NP	NP
		NO	3.34	0.80				
		HEALTHY	3.34	0.90				
EI	EIP	IN	3.27	0.91	2.48	*p* ≥ 0.05	NP	NP
		NO	3.15	0.90				
		HEALTHY	3.09	0.87				
	EIU	IN	2.94	0.85	3.72	*p* ≤ 0.05 ^c^	0.19 ^c^	[0.05; 0.33]
		NO	2.94	0.83				
		HEALTHY	3.10	0.81				
	EIR	IN	2.94	0.80	13.89	*p* ≤ 0.05 ^a, b, c^	0.26 ^a^ 0.48 ^b^ 0.21 ^c^	[0.10; 0.41] [0.30; 0.65] [0.06; 0.35]
		NO	3.16	0.86				
		HEALTHY	3.34	0.84				
DP	SUDIS	IN	2.53	0.70	21.06	*p* ≤ 0.05 ^a, b, c^	0.21 ^a^ 0.56 ^b^ 0.34 ^c^	[0.05; 0.36] [0.39; 0.74] [0.19; 0.48]
		NO	2.38	0.72				
		HEALTHY	2.14	0.67				

Note 1. Inactive (IN); engages in non-beneficial levels of physical activity (NO); engages in beneficial levels of physical activity (HEALTHY). Note 2. Overall resilience (SURE); locus of control and commitment (LCC); challenge of action-oriented behavior (CAOB); self-efficacy and resistance to discomfort (SRD); optimism and adaptation to stressful situations (OASS); spirituality (ES); emotional perception (EIP); emotional understanding (EIU); emotional regulation (EIR); overall psychological distress (SUDIS). Note 3. Differences between IN and NO (a); differences between IN and HEALTHY (b); differences between NO and HEALTHY (c). No statistically significant differences “*p* ≥ 0.05”.

**Table 3 ijerph-17-04474-t003:** Correlation between the dimensions of resilience, emotional intelligence, and psychological distress, as a function of physical activity engagement (beneficial physical activity, non-beneficial physical activity and inactivity).

Variables		NO Minimum PA (N = 621)									
YES Minimum PA (N = 474)	**Category**	**SURE**	**LCC**	**CAOB**	**SRD**	**OASS**	**ES**	**EIP**	**EIU**	**EIR**	**SUDIS**
	SURE	1	0.92 **	0.64 **	0.96 **	0.88 **	0.66 **	0.20 **	0.36 **	0.44 **	−0.35 **
	LCC	0.92 **	1	0.53 **	0.87 **	0.77 **	0.53 **	0.07	0.38 **	0.42 **	−0.29 **
	CAOB	0.60 **	0.46 **	1	0.56 **	0.49 **	0.36 **	0.09 *	0.27 **	0.28 **	−0.23 **
	SRD	0.96 **	0.87 **	0.50 **	1	0.81 **	0.57 **	0.09 *	0.31 **	0.40 **	−0.22 **
	OASS	0.89 **	0.77 **	0.50 **	0.82 **	1	0.50 **	0.06	0.34 **	0.44 **	−0,22 **
	ES	0.64 **	0.52 **	0.31 **	0.52 **	0.48 **	1	0.16 **	0.17 **	0.28 **	−0.04
	EP	0.03	−0.00	0.03	0.04	0.00	0.11 *	1	0.40 **	0.23 **	0.43 **
	EU	0.40 **	0.32 **	0.22 **	0.27 **	0.28 **	0.11 *	0.30 **	1	0.53 **	−0.12 **
	ER	0.33 **	0.32 **	0.15 **	0.31 **	0.35 **	0.17 **	0.32 **	0.54 **	1	−0.15 **
	SUDIS	−0.19 **	−0.23 **	−0.18 **	−0.17 **	−0.17 **	−0.02	0.34 **	−0.28 **	−0.24 **	1

Note 1. YES, meets minimum physical activity recommendations, and; NO, does not meet recommended minimum physical activity recommendations. Note 2. Overall resilience (SURE); locus of control and commitment (LCC); challenge of action-oriented behavior (CAOB); self-efficacy and resistance to discomfort (SRD); optimism and adaptation to stressful situations (OASS); spirituality (ES); emotional perception (EIP); emotional understanding (EIU); emotional regulation (EIR); overall psychological distress (SUDIS). Note 3. Significant correlation at the 0.05 level (*); significant correlation at the 0.01 level (**).

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
