# Peer review of "Benefits of Physical Activity and Its Associations with Resilience, Emotional Intelligence, and Psychological Distress in University Students from Southern Spain"

_ijerph, 2020, doi:10.3390/ijerph17124474_

Round 1

Reviewer 1 Report

The paper seems better than before.

Author Response

Dear reviewer,

we would like to thank you for the time spent reviewing the manuscript and your comments about it. We think the same,  the paper seems better than before because your suggestions have helped improve it, again, thank you

Reviewer 2 Report

The overall quality of the study improved significantly and most of the comments were addressed accordingly. However, the study still has important limitations that should be considered. The two major points are related with the methods used to evaluate the current population.

  1. It is still not clear the rationale to use two different metrics categorize physical activity. As it is presented now, it seems that the authors were not sure about the best metric and decided to use both. What the scale ‘reach the minimum physical activity’ brings in terms of new information? As the authors point out in the discussion, line 285: “Meeting minimum physical activity recommendations is not the same as engaging in healthy physical activity (300 minutes per week)”. But why? It would be beneficial to present the reasoning to use both scales. If the scale presented by WHO does not promote health benefits, what is the purpose of using it in this study then?
  2. The self-reported physical activity is still unclear and should be better discussed in the limitation section. Did you simply asked: “how many minutes of moderate or intense physical activity do you engage each week?” (line 101). Have you asked the participants about the type of activity or just the duration? Some people consider walking light physical activity and other moderate physical activity. Have you given any guidelines to help the participants count their minutes of physical activity? As it is presented in the limitation section, seems disconnected from the rest of the sentence.

Just a small detail, change the last paragraph of the introduction to the past tense (‘this study aimed at uncover….’ and so on).

Author Response

Dear reviewer, thanks you for your time and your contribution to improve and clarify the purpose of this manuscript, our manuscript has been reviewed by a native scientist and we hope that with the contributions made, it has been corrected.

  1. We have expanded  and explained in our introduction section, that World Health Organization (WHO) physical activity recommendations  for adults aged from 18 to 64 years of age, suggest a minimum of 150 minutes per week of moderate activity .There is a direct relationship between physical activity and cardiorespiratory health, but the risk reduction is achieved from 150 minutes of moderate or intense exercise a week. In cases where physical activity engagement increases to 300 minutes a week or above, reports additional health benefits (Line 46-49).
  2. we have explained in our variables and instruments the moderate physical activity. There is a question about relative personal ability scales with scores of 0 (does not engage in physical activity or sport) to 10 (above possibilities). The scores between 3-4  means a moderate physical activity, following the scores of Borg Scale  (Line 103-107).
  3. We have expanded de limitations with more references and the  bibliopraphy (Lines 334)
  4. We have changed a past sented the last paragraph of the introduction
  5. Thank you again, hope that with the contributions made, it has been corrected.

Round 2

Reviewer 2 Report

Changes were made based on previous suggestions.

This manuscript is a resubmission of an earlier submission. The following is a list of the peer review reports and author responses from that submission.

Round 1

Reviewer 1 Report

      It is interesting research in general that addresses a topic of health relevance. Always consider the limitations of this type of design for not establishing causality relationships.   A multivariate predictive model would further enhance the conclusions. As well, analyze gender differs and age strata differences.   In general it is a good job that can serve as the basis for future research with more complex designs

Points of weakness

About the sample

The sample has a fairly wide age range (from 17 to 54 years) and its distribution is not sufficiently shown, i.e. it would be desirable to show an age distribution plot, which allows to appreciate the asymmetry of the curve.

The sample could be better described if you add information about the specialty that the person is studied and know if it relates to studies of Health Sciences, or Physical Activity and Sports, etc.

The inclusion and exclusion criteria have not been sufficiently specified to participate in the sample. There may be important exclusion criteria such as some physical or mental illness or disorder, as well as certain special conditions.

The sample is limited to the scope of the University, but we do not know whether its external validity would allow the results to be extrapolated to non-university people.

About comparisons and analisys:

Comparisons by age group and gender should be shown, not just raw comparisons.

Comparisons made could be improved and more information provided if you consider the possible role that the gender variable can play as a confounding factor or the age variable as a possible variable modifying the effect of physical exercise on the psychological variables considered in the study.

The use of others statistical analysis (For example, Mantel and Haenszel analysis) would allow to assess the intensity of association between different age strata and assess whether that variable modifies the effect. That is, perform statistical analyses that allow to detect as relevant variables such as sex and age are or are not influencing the associations and correlations established.

Points of strength, 

Study overview

It is a study that highlights the need to promote the practice of physical exercise for the mental health and psychological well-being of people. Something that may be common sense, but that this study verifies with an extensive sample large enough.

It helps to better understand the role of Protector Factor that plays this variable in important psychological parameters such as resilience and emotional intelligence.

About measuring instruments:

The tests you use are validated tests with good proven Reliability and Validity values.

About comparison groups:

The criteria use to make cross-group comparisons are based on WHO recommendations.

About  Discussion and Conclusions:

The discussion of the results is clearly stated and corresponds to the advantages and limitations of the statistical tests used, not overstepped by the interpretation of the results. It does not fall into the error of establishing causal relationships.

The findings relate results consistently with a good range of previous studies.

Reviewer 2 Report

This paper focuses on the correlations between physical activity's practice and resilience, emotional intelligence and psychological distress in a population of around 1000 students in Andalucia. 

The paper quality is not bad, but it need a very important english editing, because some parts are not understandable. As for the litterature review (introduction), the authors quote some spanish-speaking papers, that are not accessible to all the potential readers of the paper. I suggest to quote and put in the bibliography only the non-english papers focusing on specific topics or populations that are not available in english. Similarly, I suggest to self-cite only if it's important to understand the evolution of a wider research process ("as we higlighted in our previous research ... ... taking into consideration these previous results, now we're going to focus in...")

Ok for me the methods.

As for the results and discussion, I suggest to consider also - even if in a smaller part of the paper - the specific context of Andalucia, and the fact that the research was - as I understand - carried in many universities in the region. Which are these universities? Which differences between the students in the different universities?

Reviewer 3 Report

The aim of this study was to know the frequency of the practice physical activity and the possible relationships with psychosocial factors and perceived psychological discomfort among 1095 participants. This is an important and relevant research topic, but some methodological issues impair the general quality of the study as it is. In general, it is not clear the rationale to use two different metrics categorize physical activity (one based on WHO, which divides the participants into two categories and another metric – no reference – which divides the participants into 3 categories). Besides, the method to evaluate physical activity has important limitations that were not acknowledged or addressed. Previous studies already showed that self-reported levels of physical activity are prone to bias and that participants report higher and more physical activity in self-reported measures when compared to objective measures (Gupta et al., 2018; Hallman et al., 2019; Prince et al., 2008).

The manuscript has typos and editing issues, which impairs the general understanding of the study. A language technical editing is desired.

Some specific comments are addressed below:

Abstract

  • The result section is a bit hard to grasp. The sentence is too long, please consider reviewing it. The three cases mentioned are not described in the method section, so the reader cannot understand it. Beneficial physical activity is considered to be moderate activity from 150-300 minutes/week. More than that is non-beneficial? Also, please provide the p values here and if this difference is related to all evaluated psychosocial factors or not.
  • How many participants and in which conditions they were evaluated?
  • What is the conclusion of this study?
  • Line 13: ‘mínimum’?
  • Line 15: ‘to’ duplicated
  • Line 19: recorder? Psysical?
  • Extra spaces between words in some sections

Introduction

  • The connection between physical activity and psychosocial factors is not defined in the introduction section, so the reader cannot understand why the authors evaluate this association in general. Around line 49, the authors introduce the psychosocial factors, but it is not clear its association with physical activity.
  • Line 34: please explain the term ‘favors social’
  • Even though WHO is a widely known term, please define it when using for the first time
  • Line 45: weird sentence ‘where more than half of the university students studied fail to develop the minimum recommended daily physical activity’
  • The aim presents the term ‘psychological discomfort’, but the title refers to ‘psychological distress’. Please choose one term and follow through the entire manuscript.
  • It seems misleading the aim of knowing the frequency of the practice of physical activity as healthy and beneficial habit. It was not mentioned in the methods, which health measures were accessed to assure a certain frequency of physical activity as beneficial. I suggest to rephrase it.

Methods

  • more information about the population is desired. Knowing that several aspects can affect both physical activity and psychosocial factors (BMI; use of tobacco, alcohol and other drugs; stress, anxiety and depression; number of children…), it is important to describe the population and be aware of potential confounders when doing the analysis.
  • Please provide the question asked about physical activity. Previous studies already showed that self-reported levels of physical activity are prone to bias and that participants report higher and more physical activity in self-reported measures when compared to objective measures. Also, what is the method used to determine the level of physical activity. How an activity was considerate to be moderate or light or intense?
  • It is not clear the metrics for categorizing the levels of physical activity. From what I understood, you divided the participants into inactive (no practice of physical –sports activity), physical activity not beneficial to health (≤300 minutes of moderate physical activity per week), physical activity beneficial to health (≥300 minutes of moderate physical activity per week) – please included reference. So what is the purpose of the previous sentence and the cut point of 150 minutes?
  • Line 92: strange sentence
  • Please define EIP, EIU, EIR
  • Line 188: please use SD when referring to standard deviation
  • Line 122: ass?
  • Line 124: please provide the information about the software developer (city, state)

Results

  • In general, I suggest reducing this section, using the same format of table for all tables and reviewing the terms and abbreviations used.
  • Lines 128 to 138: this information is already of the table. I suggest to reduce and highlight just the essential information. The same is truth for line 144 to 175.
  • Table 1: Provide the description for each abbreviation as a footnote. Also, please decrease the decimal cases, just two are enough. Decimals cases should also be adjusted on Table 2.
  • Terms and abbreviations should be reviewed. Resilience and SURE refer to the same metric, from what I understood. In the text the term EIP is used, but no such term can be found in the table…
  • Table2: RE, EI, DP? The way the significance is shown is confusing (a,b,c). Another format to present the p values may be desirable.
  • It’s not clear why knowing the correlation between the psychosocial factors is relevant in this study. In the title and to some extent the abstract and introduction point out to evaluating the association between physical activity and psychosocial factors.
  • Table 3: strange format. Please provide the meaning of * and **

Discussion

  • Lines 243 to 247: here is an important justification to use both measures. I suggest to briefly mention it in the introduction section and explore it more deeply in the discussion, so it would create a rationale, and the reader would understand the reason the authors had to use both categories of physical activity.

References

Gupta, N., Heiden, M., Mathiassen, S.E., Holtermann, A., 2018. Is self-reported time spent sedentary and in physical activity differentially biased by age, gender, body mass index, and low-back pain? Scand. J. Work. Environ. Heal. 44, 163–170. https://doi.org/10.5271/sjweh.3693

Hallman, D.M., Mathiassen, S.E., van der Beek, A.J., Jackson, J.A., Coenen, P., 2019. Calibration of Self-Reported Time Spent Sitting, Standing and Walking among Office Workers: A Compositional Data Analysis. Int. J. Environ. Res. Public Health 16. https://doi.org/10.3390/ijerph16173111

Prince, S.A., Adamo, K.B., Hamel, M.E., Hardt, J., Connor Gorber, S., Tremblay, M., 2008. A comparison of direct versus self-report measures for assessing physical activity in adults: A systematic review. Int. J. Behav. Nutr. Phys. Act. 5. https://doi.org/10.1186/1479-5868-5-56